# LINGUISTICALLY-GROUNDED AND EXPLAINABLE DEMONSTRATION RETRIEVAL FOR IN-CONTEXT LEARNING

## ABSTRACT

In-context learning (ICL) is an emerging ability of language models, and its effectiveness hinges on the selection of effective in-context examples for every query. Existing research predominantly rely on retrieval techniques to curate such potential examples for each query. These examples are then ranked by a specialized scoring language model, distinguishing between positive (effective) and negative (ineffective) examples as demonstrations. These results then inform the training of a dense retriever to select effective demonstrations for queries at test time. Existing approaches suffers from narrow selection criteria, lack of explainability, and limited robustness and transferability. This paper introduces a novel approach, grounded in linguistic principles, which defines the key criteria that effective demonstrations should meet. These criteria are language model agnostic, demonstrate superior performance not only in a standard ICL setting but also in domain adaptation settings and in contexts devoid of task-specific instructions, provide explanations for selecting demonstrations, and shed light on inherent biases in existing methods. The proposed approach outperforms five strong baselines across seven tasks. Notably, it achieves higher performance than explicitly optimized models for ICL, such as MetaICL, highlighting its potential applications on large scale models.

## 1 INTRODUCTION

As the scale of language models (LMs) inflates, in-context learning (ICL) emerges as a new ability of LMs. In ICL, a handful of labeled training examples (or demonstrations) and a single test example (or query) are concatenated to prompt the LM to infer the label of the query. Given the context size limitation of LMs, only a select few examples can serve as demonstrations for each query. Existing research show that the quality and formatting of demonstrations largely influence the ICL performance, and proposed techniques for demonstration selection (Li et al., 2023; Levy et al., 2023; Zhang et al., 2022; Rubin et al., 2022; Liu et al., 2022) and ordering (Zhang et al., 2022; Lu et al., 2022). The present work focuses on demonstration selection, aiming to select the most informative and performant demonstrations from labeled examples for strong and stable ICL performance.

Most existing approaches formulate the demonstration selection task as a retrieval problem, where for each query, $k$ demonstrations are selected from labeled examples based on their similarity to the query. Some methods adopt heuristic-based approaches, including L2 distance in embedding space (Liu et al., 2022) or syntactic overlap (Levy et al., 2023). Other approaches pre-select and label a few seed examples to train a dense retriever (Rubin et al., 2022; Li et al., 2023), or train a policy network with reinforcement learning (Zhang et al., 2022) for demonstration selection.

Existing methods suffer from three shortcomings: (1): *narrow selection criteria*: current heuristic-based approaches focus on particular aspects of demonstrations (e.g., syntax), and dense retriever-based methods primarily rely on lexical similarity and may be insufficient for tasks that require deeper features or reasoning (Rubin & Berant, 2021). Recent studies like (Min et al., 2022b; Yoo et al., 2022) highlight how different factors of demonstrations, e.g. the distribution of inputs and use of gold labels, can influence ICL performance. (2): *Lack of explainability*: existing methods, especially those based on dense retrieval and reinforcement learning, do not provide human-

comprehensible justifications to rationalize their selection process or insights into which demonstration factors contribute to their selection. This opacity can hinder adaptability to new tasks and generalizability of results. (3): *Compromised robustness and transferability*: existing dense retrieval-based methods may be sensitive to the choice of scoring LM, and the selected demonstrations may not seamlessly transfer to another inference LM.

In this paper, we present **L**inguistically-**G**rounded and **E**xplainable **D**emonstration **R**etrieval (LGEDR) for In-Context Learning. We propose several linguistically-grounded and human-understandable criteria that effective demonstrations and prompts need to fulfill. Based on these criteria, we design two retrieval methods to select demonstrations without the need for task-specific instructions. Critically, our approach can quantitatively explain which criteria and to which extent contribute to demonstration selection for each query. This capability not only enhances transparency but also uncovers different inductive biases of existing demonstration retrieval methods.

Experiments show that, compared to five strong baselines, the proposed approach yields an average improvement of 9.7 absolute points (in terms of F1 score and accuracy metrics) across seven NLP tasks. Notably, our method outperforms MetaICL (Min et al., 2022a), which explicitly trains large LMs (LLMs) for ICL, without incurring the significant costs of learning from hundreds of NLP datasets and extensive training.

Our main contributions include

- enriching existing demonstration selection approaches with linguistically-grounded and human-understandable criteria that effective demonstrations and prompts need to fulfill;
- showing the efficacy of the proposed demonstration selection criteria in case of smaller LMs, in the absence of task instructions, and in transfer settings; and
- providing explanations for demonstration selection from linguistic perspective.

## 2 RELATED WORK

**Instability and bias in ICL**   ICL suffers from unstable performance. With different sets of demonstrations, ICL performance varies and its is not guaranteed to improve when more demonstrations are added or when larger models are used (Zhao et al., 2021; Zhang et al., 2022). Different ordering of demonstrations can lead to variant performance (Zhang et al., 2022; Lu et al., 2022). Zhao et al. (2021) find that GPT-3's predictions are influenced by majority label, recency, and common token biases in demonstrations. Majority label bias indicates a skew toward frequently appearing labels in demonstrations; the more often a label is present, the more likely the model is to produce it. Recency bias indicates bias toward labels of demonstrations that closest to the queries in the prompt. Common token bias indicates bias toward common tokens in the pre-training corpus. Chen et al. (2023) find that the sensitivity of a prompt is negatively correlated with its performance. In addition to demonstrations, the instruction wording can cause the LM to flip its predictions, even in cases where both instructions are semantically similar to human beings (Chen et al., 2023). The overall format of the prompt can also have a strong impact on the ICL performance. Min et al. (2022a) find that replacing the input and the label for each demonstration used can lead to better performance. To stabilize ICL, Zhao et al. (2021) developed techniques that impose LLMs to assign equal probability across the outputs in case of context-free prompts, and show that such calibration results in a more stable performance. Chang & Jia (2023) find a subset of examples based on development set performance from which arbitrary selection leads to low standard deviation of performance.

**Explanation of demonstration**   Recently, several work have investigated different aspects of demonstrations affects ICL performance. Min et al. (2022b) observe that replacing ground-truth labels in demonstrations barely hurts the performance on several classification and multi-choice tasks, while the label set has a larger influence. On the other hand, Yoo et al. (2022) argue that the impact of ground-truth labels varies across different experimental setups. Wei et al. (2023) show that only LLMs have such flexibility to override their prior knowledge and learn from semantically unrelated labels. Ye et al. (2023) show that a diverse demonstration set can result in a better reasoning performance, and computation trace and wording of the explanations can better express the prompt.

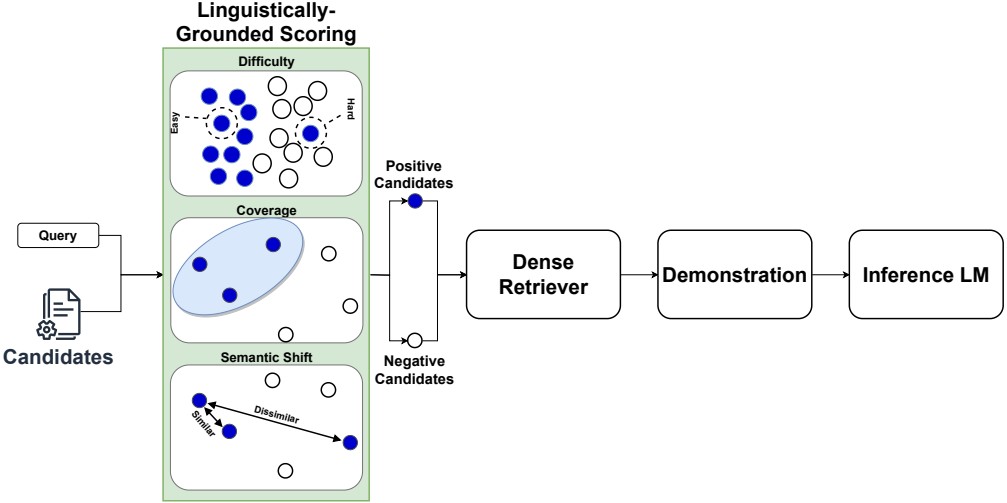

Figure 1: Illustration of LGEDR. We propose three linguistically-inspired criteria to measure the quality and rank candidates.

**Demonstration selection**    Previous works mostly formulate demonstration selection as a retrieval problem, where a set of demonstrations is selected from a large labeled set (usually the training set of the task), either for a specific text example or for all test examples. Rubin et al. (2022) trains a dense retriever to select effective demonstrations, where the training data is generated by scoring each pair of training examples with a scoring LM. However, examples scored positively by the scoring LM across training pairs does not necessarily indicate their effectiveness in the downstream inference LM on the test set. Li et al. (2023) iteratively score candidates and train a unified dense retriever for various NLP tasks with task instructions. However, the performance improvement depends largely on iterative scoring and training, and the quality of task instructions, which requires domain knowledge. Besides dense retrievers, heuristic-based metrics are proposed, such as L2 distance (Liu et al., 2022) and diversity of syntactic structure (Levy et al., 2023). Zhang et al. (2022) formulates the problem as a sequence of decisions and applies reinforcement learning to determine the optimal choice of demonstrations at each step. Although the approach introduces a new perspective, it can be expensive since the action space is exponential with respect to the size of the candidates. Chang & Jia (2023) score each training example based on its performance on validation set, resulting in a stable set of examples, where random selection from the set yields good performance. Existing works are limited by lexical-based candidate filtering and inefficient LM-based candidate scoring, which is the focus of the present work.

## 3    PROPOSED APPROACH

**Problem formulation**    Given a set of labeled candidates $\mathcal{C}$, a test set $\mathcal{Q}$, and an inference language model $f$, we aim to train a dense retriever that, for every test query $q \in \mathcal{Q}$, selects $k$ query-specific training examples from $\mathcal{C}$ and constructs them as a prompt $\mathcal{P}$ such that $f(\mathcal{P})$ correctly produces the label of the test query $q$.

### 3.1    LINGUISTICALLY-GROUNDED DEMONSTRATION FEATURES

In order to select effective demonstrations, we advocate for the adoption of distinct criteria, as illustrated in Figure 1. Unlike existing techniques that rely on a single scoring LM to access the quality of demonstrations, our approach integrates several linguistically-grounded criteria for a more comprehensive evaluation. Adopting these criteria results in several advantages over a single scoring LM. First, they augment the traditional scoring LM-based selection with intuitive and human-understandable attributes. Second, the need for a scoring LM, thus the associated costs, can be avoided. Third, being independent of any specific scoring LM, our criteria result in better robustness and transferability compared to sole reliance on a single scoring LM.

**Difficulty**  We hypothesize that a good set of demonstrations should match the difficulty level of the query, necessitating similar extents of knowledge or reasoning required to predict the label for the query. Easy demonstrations may not provide sufficient information for hard queries, whereas hard demonstrations may be excessive for easy queries. Drawing inspiration from previous work on neural density estimation and curriculum learning (De Cao et al., 2020; Wang et al., 2021), we characterize the difficulty of an example through the class imbalance of its neighboring examples, regularized by the number of such neighbors. Specifically, for each test example $x_i$, we first define its immediate neighbors $\mathcal{N}_i$ as the training examples lying within a spherical space of radius $r$ in an embedding space:

$$\mathcal{N}_{i,r} = \{j \big| \|x_j - x_i\| \leq r, \forall x_j \in \mathcal{C} \backslash x_i\}, \tag{1}$$

where $\mathcal{C}$ is set of training examples; in what follows, we omit the subscript $r$ from $\mathcal{N}_{i,r}$ for simplicity. As illustrated in Figure 1, if $y_i$ is the same as the label of most neighbors $y_{\mathcal{N}_i} = \{y_j, j \in \mathcal{N}_i\}$, then $x_i$ is deemed easy because its class can be easily inferred from its neighboring examples. However, if $y_i$ is different from the label of most neighbors, then $x_i$ is likely to be a hard example. The difficulty of $x_i$ can thus be measured through the class imbalance within $\mathcal{N}_i$. We compute the number of neighbors that share the same label as $x_i$ as:

$$N|_{y_{\mathcal{N}_i}=y_i} = \sum_{j \in \mathcal{N}_i} \mathbf{1}_{y_j=y_i} \tag{2}$$

where $\mathbf{1}$ is the indicator function. The difficulty of $x_i$ can be expressed as its class imbalance:

$$\text{Diff}(x_i) = 1 - \frac{N|_{y_i=y_{\mathcal{N}_i}}}{|\mathcal{N}_i| + \epsilon}, \tag{3}$$

where $\epsilon$ is used to regularize the number of neighboring examples of $x_i$, which varies considerably across training examples and needs to be taken into account for difficulty measurement. Examples with few neighbors tend to be difficult due to lack of sufficient information about them. Given the maximum and minimum neighbor counts for training examples, we define a regularization term $\epsilon_i$ for each $x_i$ using the following geometric mean:

$$\epsilon = \left( \frac{1}{|\mathcal{C}|} \max_j |\mathcal{N}_j| \right)^{(1-\gamma)} \times \left( \frac{1}{|\mathcal{C}|} \min_j |\mathcal{N}_j| \right)^{\gamma}, \tag{4}$$

where $\gamma \in [0, 1]$ is a hyperparameter to weight the maximum and minimum number of neighbors.

**Coverage**  We hypothesize that a good set of demonstrations should cover the breadth and diversity of the training examples, thereby enabling the LM to reconstruct the latent concepts and underlying nuances of the labels. A higher coverage is more likely to describe the classes and their boundaries to the LM in a more systematic manner. To quantify this coverage, we use the minimum area of a surface (an circle or ellipse) in the embedding space. For a set of $k$ demonstrations, we can compute their coverage as:

$$\text{Cov}(\{x_i\}_{i=0}^{k-1}) = \min_{M \in \mathcal{M}} \oint M(\{F(x_i)\}_{i=0}^{k-1}), \tag{5}$$

where $F$ denotes a pre-trained feature extractor and $M$ denotes the manifold on which the $k$ demonstrations reside.

**Semantic shift**  We hypothesize that a good set of demonstrations should exhibit a cohesive and smooth semantic relationship, both among themselves and in relation to the query. Such similarity aids in collectively steering the LM toward a correct prediction. Moreover, semantic consistency within a prompt can help determine the predictability of the next token. We measure this consistency by quantifying semantic shift between successive demonstrations and between demonstrations and the query. Given two examples $x_i$ and $x_j$, we use the L2 distance between their sentence embeddings as a proxy of semantic shift:

$$\text{Shift}(x_i, x_j) = \|F(x_i) - F(x_j)\|. \tag{6}$$

## 3.2 DEMONSTRATION RETRIEVAL

For each query, we rank the candidates with the above three linguistic criteria, and average the relative order to obtain a final ranked list of candidates, where the top and the bottom examples are labeled as positive and negative examples for the query respectively. Then, we train a dense retriever (Karpukhin et al., 2020) to select demonstrations for an input query using a contrastive loss. Specifically, for each query $q_i$, we sample one positive example $x_i^+$ and several negative examples $\{x_0^-, \ldots, x_{N_{neg}-1}^-\}$ and optimize the following contrastive loss:

$$L = -\log \frac{e^{\text{sim}(q_i, x^+)}}{\text{sim}(q_i, x^+) + \sum_j \text{sim}(q_i, x_j^+)}, \tag{7}$$

where $\text{sim}(x_i, x_j) = E(x_i)^T E(x_j)$ and $E$ is the encoder of the dense retriever. At inference time, we propose two retrieval methods to use the trained retriever and select the $k$ demonstrations.

### 3.2.1 BATCH AND SEQUENTIAL RETRIEVAL

As shown in Algorithm 1, given a query $q$ for prediction, batch retrieval selects $k$ demonstrations directly in a single step. The retrieved demonstrations are ordered according to the inverse ranking provided by the retriever. On the other hand, as shown in Algorithm 2, sequential retrieval selects *one* demonstration at a time for a given query. After selection, the most recently selected demonstration is treated as the new query for retrieval. The demonstrations are ordered according to the reverse order that they are retrieved.

Batch retrieval provides a holistic view, giving the LM all demonstrations at once, while sequential retrieval provides an incremental view. This incremental approach might allow the LM to build upon each piece of information step by step, making the impact of the above criteria more pronounced

---

**Algorithm 1** Batch retrieval

**Require:** Query $q$, labeled candidates $\mathcal{C}$, trained retriever $\mathcal{R}$, number of demonstrations to select $k$
**Ensure:** $\mathcal{P}$        ▷ prompt
1: $\mathcal{P} \leftarrow \emptyset$
2: $\mathcal{C}_{ranked} \leftarrow \mathcal{R}(q, \mathcal{C})$    ▷ rank candidates
3: **for** $i$ in 1 to $k$ **do**
4:      $d_i \leftarrow \mathcal{C}_{ranked}[i]$   ▷ closest candidate
5:      $\mathcal{P} \leftarrow d_i \oplus \mathcal{P}$      ▷ prepend $d_i$ to $\mathcal{P}$
6: **end for**

---

**Algorithm 2** Sequential retrieval

**Require:** Query $q$, labeled candidates $\mathcal{C}$, trained retriever $\mathcal{R}$, number of demonstrations to select $k$
**Ensure:** $\mathcal{P}$        ▷ prompt
1: $\mathcal{P} \leftarrow \emptyset$
2: **for** $i$ in 1 to $k$ **do**
3:      $\mathcal{C}_{ranked} \leftarrow \mathcal{R}(q, \mathcal{C})$
4:      $d_i \leftarrow \mathcal{C}_{ranked}[0]$ ▷ closest candidate
5:      $\mathcal{C} \leftarrow \mathcal{C} \backslash d_i$        ▷ remove $d_i$
6:      $\mathcal{P} \leftarrow d_i \oplus \mathcal{P}$     ▷ prepend $d_i$ to $\mathcal{P}$
7:      $q \leftarrow d_i$    ▷ set current query to $d_i$
8: **end for**

---

## 4 EXPERIMENTAL SETUP

**Setup** Following previous work, we adopt a minimal prompt template in which input and its label are simply concatenated and no auxiliary words or task instructions are added. We evaluate each method with $k = 4$ similar to previous work (Zhang et al., 2022), using on GPT-2 (Radford et al., 2019) and GPT-J-6B (Wang & Komatsuzaki, 2021) models. For LGEDR, we apply the retriever in sequential and batch manner, denoted as LGEDR-Seq and LGEDR-Batch, respectively.

**Dataset** We evaluate our method on a range of NLP tasks and datasets, leveraging datasets commonly cited in prior studies. For sentiment analysis, we test on SST2 (Socher et al., 2013) and Rotten Tomatoes (Pang & Lee, 2005). For natural language inference, we use RTE (Wang et al., 2019) and WNLI (Wang et al., 2019). For paraphrase identification, we use MRPC (Dolan & Brockett, 2005). For linguistic acceptability, we use CoLA (Warstadt et al., 2019).

Table 1: In-context learning performance on all test sets with different demonstration selection methods on GPT-2. **Bold** and underline indicate highest and second highest performance, respectively.

| Model | SST2 | RT | CoLA | RTE | WNLI | MRPC | Trec | Average |
|---|---|---|---|---|---|---|---|---|
| Random | 62.1 | 55.8 | 59.7 | 47.6 | 55.0 | 37.8 | 18.1 | 48.0 |
| KATE | 45.9 | 53.3 | 62.8 | 41.1 | 52.4 | 31.6 | 17.3 | 43.4 |
| COVER-LS | 33.7 | 48.4 | 44.7 | 46.5 | 38.7 | 49.0 | 17.5 | 39.9 |
| EPR | 56.8 | 60.1 | 63.4 | 47.9 | 50.5 | 52.1 | 16.4 | 49.6 |
| UDR | 59.6 | 63.3 | 68.1 | 50.4 | 55.8 | 63.7 | 17.6 | 54.1 |
| AES | **71.3** | 54.7 | 52.5 | 43.2 | 35.7 | 38.6 | 40.1 | 48.1 |
| **LGEDR-Batch (ours)** | 68.2 | **65.3** | **68.8** | 52.3 | **57.7** | 63.8 | 39.2 | 59.3 |
| **LGEDR-Seq (ours)** | 67.3 | 62.4 | 67.3 | **56.3** | 57.1 | **68.3** | **40.2** | **59.8** |

**Baseline**  We compare LGEDR with the following methods:

- **Random** (Liu et al., 2022) (**heuristic-based**): randomly selects $k$ examples from training data $\mathcal{C}$ as demonstrations.

- **KATE** (Liu et al., 2022) (**heuristic-based**): encodes $q$ and $\mathcal{C}$ as embeddings and retrieves $k$ nearest neighbors of $q$ based on L2 distance as demonstrations.

- **COVER-LS** (Levy et al., 2023) (**heuristic-based**): selects a subset of examples that collectively achieve diverse syntactic structure of input utterances as demonstrations.

- **EPR** (Rubin et al., 2022) (**dense retriever-based**): selects potentially useful candidates using a traditional retrieval model (BM25), concatenates each candidate with the query and scores it with a scoring LM, where the candidates ranked top are treated as positive samples and the candidates ranked bottom are treated as negative samples. A dense retriever is trained with the data and used to retrieve demonstrations at test time.

- **UDR** (Li et al., 2023) (**dense retriever-based**): iteratively retrieves candidates similar to task instructions and trains the dense retriever. To make a fair comparison with our method, we only train it once.

- **ASE** (Zhang et al., 2022) (**reinforcement learning-based**): trains a reinforcement learning policy to select demonstrations from a pool of candidates and label them.

## 5 RESULTS

**LGEDR selects informative demonstrations**  Table 1 compares our method with baselines on several datasets. On average across all datasets, LGEDR outperforms random selection by 11.6 absolute points. Compared to dense retrieval approaches, LGEDR outperforms EPR and UDR by 10.0 and 5.4 absolute points respectively. Compared to heuristic-based approaches, it outperforms KATE and COVER-LS by 16.1 and 19.6 absolute points respectively. It also outperforms AES by 17.7. The mediocre performances of EPR, UDR, and COVER-LS can be attributed to their dependency on lexical-based methods to retrieve demonstrations. EPR and UDR adopt BM25 to label relevant candidates for training the dense retriever and COVER-LS computes lexical diversity. Lexical overlap or diversity may be a good indicator of the quality of training examples on natural language utterance datasets, but it may not be a good indicator on tasks like NLI that require sophisticated reasoning capability. In fact, as many previous works (Karimi Mahabadi et al., 2020; Gao et al., 2022) point out, lexical overlap leads to dataset biases and can be misleading for NLI tasks. In addition, LGEDR outperforms two heuristic-based approaches KATE and COVER-LS by 16.2 and 19.7 points respectively. These results indicate that a single heuristic may not always select effective demonstrations, which depends on both the dataset and the inference LM. For UDR, we notice that its performance drops significantly compared to the performance reported in the original paper (Li et al., 2023). This may be due to (a): no iterative scoring plus retriever training and (b): different scoring and inference LMs. We argue that having access to the downstream inference LM and applying it to score a large number of candidates is not a practically applicable setting. This comparison highlights the advantages of LGEDR because it does not depend on a scoring LM to examine the demonstration quality.

Table 2: Ablation study on linguistic criteria for demonstration selection.

| Model | Full model | W/o Difficulty | W/o Coverage | W/o Semantic Shift |
|---|---|---|---|---|
| **LGEDR-Batch** | **59.3** | 55.6 (-3.7) | 56.3 (-3.0) | 55.2 (-4.1) |
| **LGEDR-Seq** | **59.8** | 54.7 (-5.1) | 56.5 (-3.3) | 55.3 (-4.5) |

**All linguistic criteria have contributions**    Through ablation analysis, we find out that all the linguistic criteria we developed for demonstration selection significantly influence downstream ICL performance. On average, removing semantic shift, coverage, and difficulty criteria lead to performance drop of 4.3, 3.2 and 4.4 absolute points respectively, see Table 2. The decline in performance for LGEDR-Batch is smaller than LGEDR-Seq, suggesting that sequential retrieval benefits more from the proposed criteria than batch retrieval. We attribute this to the iterative and incremental nature of sequential retrieval, which makes it more sensitive to the quality and relevance of demonstrations as any misstep in one selection can cause subsequent retrievals to deviate from the optimal path. Among the linguistic criteria, coverage appears to be the least important for both retrieval methods. This is perhaps because coverage seeks to increase diversity in demonstrations and an overemphasis on diversity might introduce less relevant or misleading examples. In fact, a more focused depth on specific areas could offer better understanding than superficial breadth, especially for tasks with a narrower focus. In addition, after covering the primary concepts, expanding coverage may yield diminishing returns, as further demonstrations might not significantly enhance the model's performance. Finally, difficulty is the most important criterion for sequential retrieval, while semantic shift is most important for batch retrieval. This is perhaps because sequential retrieval selects demonstrations one at a time, making it crucial to prioritize examples based on their difficulty level to ensure a progressive and coherent prompt. On the other hand, batch retrieval selects demonstrations all at once, emphasizing the importance of minimizing semantic shifts to maintain cohesion and consistency in the prompt.

**LGEDR is good at domain adaptation**    We also evaluate LGEDR under domain adaptation setting, where the training and test sets are from different domains, for example selecting demonstrations from SST2 and making inference on Rotten Tomatoes. As Table 3 shows, LGEDR maintains robust performance in this evaluation setting, outperforming KATE and COVER-LS by 9.7 and 4.5 absolute points respectively. Among the baselines, ASE and UDR notably outperform others. We hypothesis that their strong performance is potentially due to the fact that they mix together training sets of multiple datasets for training their demonstration selector.

**LGEDR selects transferable demonstrations**    To assess whether the selected demonstrations can be readily applied to a different inference LM, we evaluate the performance of the same set of demonstrations selected by each method on GPT-J (6B) and GPT-2 (142M) and report their gap. Results in Table 4 show that using the same demonstrations, LGEDR suffers from a modest performance drop of 3.3 absolute points, which is much smaller than that of UDR, EPR, and KATE. We attribute this strong transferability to the linguistically-grounded criteria that are agnostic to both scoring LM and inference LM. It is worth noting that KATE can select informative demonstrations based on L2 distances on GPT-3, but such demonstrations do not work for GPT-2, which is a considerably smaller model.

**LGEDR is robust to the choice of feature extractor**    One of the bottlenecks of existing dense retrieval-based methods is that they rely on scoring candidates with a scoring LM. High performance on the candidates with a specific scoring LM does not necessarily indicate high performance on the test set with another inference LM. Adopting the same scoring LM as the inference LM, as in UDR (Li et al., 2023), is not practical in many cases, such as when working with close-source LLMs such as GPT-3. Appendix Table 6 presents the average performance of using three different scoring LMs and another inference LM with EPR and UDR. For LGEDR, we also use three different pretrained models to generate embeddings. The results show that LGEDR has standard deviation of 3.3 across different embedding methods, much smaller than that of EPR and UDR, 8.3 and 10.7 respectively. This result highlights the strong robustness of the proposed criteria.

Table 3: Performance under domain adaptation setting on GPT-2.

| Model | SST2 →RT | RT →SST2 | RTE →WNLI | WNLI →RTE |
|---|---|---|---|---|
| Random | 50.6 | 47.8 | 49.5 | 46.0 |
| KATE | 39.3 | 40.8 | 44.6 | 41.3 |
| COVER-LR | 40.7 | 45.3 | 50.0 | 46.5 |
| EPR | 55.7 | 51.3 | 49.5 | 46.6 |
| UDR | 61.0 | 57.2 | 51.7 | 43.8 |
| ASE | 53.3 | **64.8** | 35.1 | 42.9 |
| **LGEDR-Batch** | **61.4** | 57.6 | 52.7 | 50.4 |
| **LGEDR-Seq** | 60.3 | 58.4 | **53.5** | **51.6** |

Table 4: Performances of the same set of demonstrations on GPT-J (6B) and GPT-2 (124M).

| Model | On GPT-J (6B) | On GPT-2 (124M) | Gap |
|---|---|---|---|
| Random | 50.4 | 48.0 | 1.6 |
| KATE | 48.8 | 43.4 | 5.4 |
| COVER-LR | 43.8 | 39.9 | 3.9 |
| EPR | 56.3 | 49.6 | 6.7 |
| UDR | 60.2 | 54.1 | 6.1 |
| ASE | 52.6 | 48.1 | 4.5 |
| **LGEDR-Batch** | **64.6** | 59.3 | 5.3 |
| **LGEDR-Seq** | 63.5 | **59.8** | 3.7 |

**Batch vs. Sequential** There is not a retrieval pattern that is always better than the other one. On average, sequential retrieval outperforms batch retrieval by 0.5 absolute points, specifically on RTE, MRPC, and Trec. On the rest of the datasets, batch retrieval has a better performance. We notice that Batch is less sensitive the selection criteria, where dropping criteria leads to a larger performance degradation, see Table 2. Batch also selects demonstrations with better transferability, see Table 4.

**LGEDR is efficient** LGEDR saves 1.5 and 2.8 hours of training time compared to EPR and UDR respectively, see Table 7 in Appendix. One major improvement is that LGEDR avoids expensive LM-based pairwise candidate scoring. Instead, the linguistic criteria can be pre-computed and indexed with efficient techniques, such as FAISS (Johnson et al., 2021), for all examples. This allows the model to quickly generate data to train the retriever. ASE formulates the problem as sequential decision making, requiring the highest training time, with 6.5 hours for trajectory generation and 8.1 hours for policy training. All costs are measured on an NVIDIA A100 GPU with 42G RAM.

## 6 DISCUSSION: WHAT IS IMPORTANT FOR ICL PERFORMANCE?

**Explanations by LGEDR** The proposed linguistic criteria offer insights that can explain the selection mechanism of any demonstration selection method. Figure 2 illustrates the normalized contribution of each criteria obtained from demonstrations selected by each model. For heuristic-based approaches like KATE and COVER-LS, their respective primary heuristics, i.e. semantic shift and coverage respectively, dominate their demonstration selection process. In contrast, dense retrieval-based methods show a more balanced contribution of all linguistic criteria, which explain their better performance than heuristic-based approaches.

**Demonstration retrieval vs. explicit ICL training** In addition to demonstration selection, several previous works focus on explicitly optimizing language models with an ICL objective to increase a model's ICL ability. Examples include MetaICL (Min et al., 2022a), In-Context Tuning (Chen et al., 2022) and Pre-training for ICL (Gu et al., 2023). However, the training demand of these models is substantial. They require collecting and pre-processing hundreds of datasets (e.g., 142 datasets in case of MetaICL), significant GPU memory, and hundreds of GPU hours. Conversely, demonstration selection techniques consume far fewer resources compared to explicit optimization. Table 5 show that most demonstration retrieval methods outperforms MetaICL, with LGEDR outper-

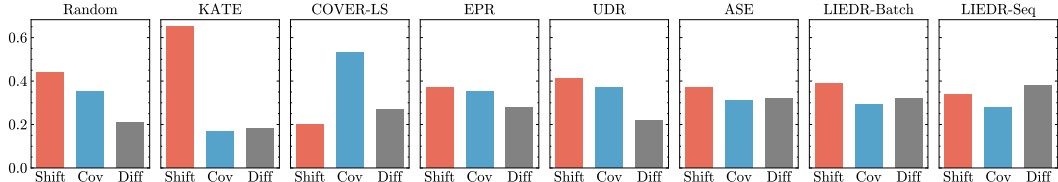

Figure 2: Explanation provided by LGEDR in terms of normalized contributions of the proposed demonstration selection criteria. For each method, we obtain the 4 demonstrations and compute the resulted difficulty, coverage and semantic shift. We normalize it so that the total contributions sum to 1.

Table 5: Comparison against the MetaICL model that is explicitly optimized for ICL. For MetaICL, we use the publicly available checkpoints for all settings where the dataset is not present in the training split. Experiments are conducted on GPT-2 Large to match MetaICL's checkpoints.

| Model | SST2 | RT | CoLA | RTE | WNLI | MRPC | Trec | Average |
|---|---|---|---|---|---|---|---|---|
| MetaICL | **76.4** | 75.7 | 57.4 | 49.5 | 53.8 | 31.6 | 22.2 | 52.9 |
| LGEDR-Seq | 71.3 | **75.8** | **68.8** | **52.7** | **60.1** | **69.1** | **40.1** | **62.9** |
| LGEDR-Batch | 71.1 | 75.2 | 67.8 | 52.1 | 59.4 | 68.3 | 39.3 | 62.3 |

forming MetaICL by 9.7 absolute points across datasets. Given its superior efficacy and efficiency, demonstration retrieval is generally more advantageous than explicit ICL training. However, explicit optimization models like MetaICL might be valuable in cases where obtaining high quality labeled examples is expensive, either for direct ICL applications or for bootstrapping demonstrations.

**Calibration does not always help** Zhao et al. (2021) reported that biases in GPT-3's predictions can be reduced using calibration with context-free inputs. However, as Figure 3 in Appendix shows, such calibration isn't universally beneficial. For smaller models (such as GPT-2), the effect on ICL performance stabilization is inconsistent and can sometimes negatively affect task performance, as seen with CoLA. This might stem from the smaller model's constrained ICL capabilities, which diminishes the overall influence of bias and calibration. For example, while calibration boosts GPT-2's performance by 4.9 absolute points on SST2, it leads to a decrease of 5.6 points on CoLA.

**Balanced labels does not always help** Several previous works have found that the label space can impact the ICL performance (Min et al., 2022b; Yoo et al., 2022). We experiment with label balanced and non-balanced demonstrations on GPT-2 and find that balanced label do not always help. Specifically, using balanced label on SST-2 results in a decrease by 0.9 absolute points. While on CoLA, using a label-balanced demonstration set can lead to an increase of 2.3 absolute points.

## 7 CONCLUSION

We introduce three linguistically-grounded and interpretable criteria to evaluate the quality of demonstrations for in-context learning (ICL). By ranking candidates, our approach eliminates the need for expensive LM-based candidate scoring while enable the training of a dense retriever for in-context demonstration selection. Experimental results demonstrate that the proposed linguistic criteria outperform both existing heuristic-driven and dense retrieval-based methods across multiple datasets and under various settings. Crucially, our proposed method–LGEDR–show greater robustness than existing methods, whose performance tends to fluctuate with changing scoring LM. Furthermore, LGEDR excels in choosing demonstrations that are more effectively transferable across various inference LMs. Finally, our approach also offers quantitative insights into the rationale behind demonstration selection methods.

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

## A APPENDIX

### A.1 ADDITIONAL RESULTS

We present the performance across three different scoring LMs in Table 6. The training time comparison is shown in Table 7. Performance comparison of calibration vs. non-calibration is shown in Figure 3.

Table 6: Average performance across three different scoring LMs.

| Model | Ave. (Std.) |
|---|---|
| Random | 46.3 $\pm$0.0 |
| KATE | 45.1 $\pm$5.1 |
| COVER-LS | 39.9 $\pm$0.0 |
| EPR | 54.2 $\pm$8.3 |
| UPR | 62.4 $\pm$10.7 |
| ASE | 52.6 $\pm$6.8 |
| LGEDR-Batch | 61.9 $\pm$3.1 |
| LGEDR-Seq | 62.8 $\pm$3.5 |

Table 7: Training time of different methods

| Model | Training time (hr) |
|---|---|
| KATE | N/A |
| COVER-LR | N/A |
| EPR | 3.8 |
| UPR | 5.1 |
| ASE | 14.6 |
| LGEDR | 2.3 |

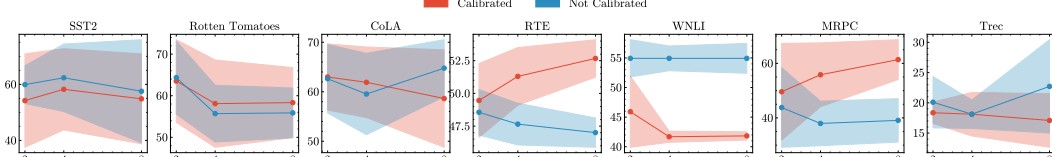

Figure 3: Performance comparison of calibration (red) vs on-calibration (blue) on GPT-2.

