# OpenReview forum: "Linguistically-Inspired and Explainable Demonstration Retrieval for In-Context Learning"
_ICLR.cc/2024/Conference — ICLR 2024 Conference Withdrawn Submission_

### Official Review · Reviewer_2Ceg · 2023-10-30

**Soundness:** 3 good
**Presentation:** 3 good
**Contribution:** 2 fair
**Rating:** 5
**Confidence:** 3

**Summary:**

This paper proposed a new method for the selection of demonstration examples. Different from the previous methods, the proposed method is based on three heuristic rules: difficulty, coverage, semantic shift. The proposed method is empirically investigated with GPT-2 and GPT-J, based on commonly used ICL datasets like SST2, RTE, WNLI.

**Strengths:**

S1. The paper is well motivated. It is important to explore effective mechanism for the selection of demonstration examples.
S2. The proposed method is empirically effective compared with baselines like ASE, UDR.

**Weaknesses:**

W1. The ICL capability of GPT-2 and GPT-J is limited. It is a major concern of the experiment result. It is necessary to perform re-investigations based on ChatGPT API or open-sourced LLMs like Llama to verify the conclusion.
W2. Several important baselines are missing, like llm-retriever [1] and uprise [2]. It necessary to make comparisons with these baselines.
W3. Baselines like llm-retriever exhibit strong experimental performances in selecting demonstration examples for ICL. It is quite interesting to check whether their selected examples satisfy the claimed three properties in this work.

[1] Learning to Retrieve In-Context Examples for Large Language Models, Wang et. al.
[2] Uprise: Universal prompt retrieval for improving zero-shot evaluation, Cheng et. al.

**Questions:**

Please check the posted weaknesses.

---

### Official Review · Reviewer_m8eZ · 2023-11-01

**Soundness:** 2 fair
**Presentation:** 2 fair
**Contribution:** 2 fair
**Rating:** 3
**Confidence:** 4

**Summary:**

This paper proposes a new method for retrieving demonstrations for in-context learning, where the retriever considers 3 linguistic principles: difficulty, coverage, and semantic shift.  The authors first describe how to measure an example based on each principle. The authors than apply the criteria on a training set, find the best and worst examples for a query,  and use them as positive and negative data to train a dense retriever. Experiments show that the the proposed model can outperform heuristic-based retrievers as well as dense-retrievers optimized for demonstration retrieval.

**Strengths:**

- The three linguistic criteria for selecting in-context learning demonstrations are novel.
- This paper studies whether demonstrations should be retrieved sequentially, which is an interesting research problem.

**Weaknesses:**

The authors doesn't provide sufficient technical descriptions, making it hard to judge the soundness of this paper. All 3 linguistic measurements are not clear. I do not understand how "coverage" is used to rank each example, since "coverage" seems to be a function of multiple examples, not one. For "semantic shift", the authors did not describe the feature extractor F in eq 6, which is critical.
It is also hard to tell how the demonstration retriever is being trained. Basic question like "what are the training data" and "what is the model" are left unanswered.

Missing these important technical descriptions, I cannot tell if the authors' claims can be justified by the experiments. A key missing piece is whether the proposed retriever is trained on in-domain data or not. If in-domain training data is used, then it is unfair to compare with the baselines which are either un-trained heuristics, or trained on diverse but out-of-domain data (EPR, UPR, MetaICL).

The proposed method would be very difficult to reproduce due to the missing technical details.

**Questions:**

- What is the training data for the retriever? What model does the dense retriever use?
- How is "coverage" computed for each individual example?
- For "difficulty", are you choosing the most difficult example, or example that has a similar difficulty level as the query?
- What is F in eq 5 and eq 6?
- It seems that the 3 criteria doesn't need learning. If so, can you directly rank examples by these 3 functions instead of training a retriever?

---

### Official Review · Reviewer_Sf1P · 2023-11-02

**Soundness:** 3 good
**Presentation:** 3 good
**Contribution:** 2 fair
**Rating:** 5
**Confidence:** 4

**Summary:**

The paper addresses the problem of finding relevant and explainable in context learning examples for language models. It proposes a 3-fold metric to rate the examples based on different linguistic criteria. These ratings are in-turn used to train a dense retriever to retrieve the right examples at run time. The method leads to better results for 4-shot learning with GPT2 on several datasets.

**Strengths:**

1. The set of ranking criteria introduced are intuitive and well motivated
2. The method improves on previous baselines on multiple datasets
3. The ranking criteria (and the resulting retriever model) is agnostic to the inference LM, and could be potentially used off-the-shelf.
4. The paper is generally well written and cites the relevant literature

**Weaknesses:**

1. I am not sure if the ideas presented in the paper are novel. Training a dense encoder to retrieve examples, as cited by the authors, has been well studied in the literature.
2. While the ranking criteria are intuitive (and do not rely on an LLM unlike some prior methods), I do not understand the utility of human understandable criteria for selecting examples. Especially since smaller models are prone to follow semantic priors present during pre-training [Wei et al. 2023]. Perhaps for larger models this might be useful for red-teaming LLMs etc. but that is outside the scope of this paper.
3. Missing details about ranking, training and comparison with other baselines (please see questions below)

**Questions:**

1. How are the candidates retrieved ranked? Say the query is $q$. For the given $q$ one calculate the difficulty say $d$. What is the exact algorithm here for ranking? From the corpus $C$, does one first select $M$ results and then calculate for each result $M_i$ a difficulty score $d_i$, semantic shift score $s_i$ and a coverage score $c_i$ ?

    1.1 What is $M$ here? Is it the entire corpus?

    1.2 How is the coverage score $c_i$ calculated for a single $M_i$? The coverage score is defined for $k$ items together. Does one need to take ${}^MC_k$ different groups?

    1.3 Once the scores are obtained how are they ranked? Are the three ranked lists (which are then averaged) obtained by sorting $d-d_i$, $s_i$ and $-c_i$ ?

     1.4 Is $s_i = || F(q) - F(q_i) || $ ?


2. What is the model size, training details of the DPR model?

3. What is the feature extractor model F? Is it a frozen pretrained DPR model?

4. Which GPT2 model is being used in Table 1?

5. Table 6 in the appendix requires a lot more attention. What are the different LGEDR models and inference models being used?

6. How are the baselines being evaluated? For instance is EPR trained on the original datasets (Break, Mtop etc.) or is trained on the training splits of SST2, RT, CoLA etc.

7. It would be helpful have to some examples/statistics of the datasets being studied in the appendix

8. [Broad question] What are the benefits of having interpretable criteria for selecting examples? Is it the belief that these criteria are robust and will work for a broad set of models and data distributions?

---

### Official Review · Reviewer_6tfm · 2023-11-02

**Soundness:** 3 good
**Presentation:** 4 excellent
**Contribution:** 3 good
**Rating:** 6
**Confidence:** 4

**Summary:**

This paper proposes a method for selecting a set of demonstrations (few-shot examples) for a given query, to improve few-shot LM quality. They train a dense retriever that maps the query to the "best" k examples. To do this, they propose difficulty, coverage, and semantic shift as three "linguistically-inspired" criteria for example selection.

In particular, they argue: (1) Demonstrations should match the difficulty level of the query. They quantify difficulty of an example through the class imbalance of its neighboring examples (easy example share the same label as most of their neighbors). Moreover, they argue that: (2) Good demonstration sets should cover the diversity of the training examples. They quantify coverage via the minimum area of a surface (an circle or ellipse) in the embedding space. Lastly, the argue that: (3) good demonstration sets exhibit a cohesive semantic relationship, both among themselves and in relation to the query, which helps determine the predictability of the next token. They measure this consistency by quantifying semantic shift between successive demonstrations and between demonstrations and the query with L2 distance of sentence embeddings.

Their method can quantitatively explain which criteria contribute to demonstration selection and uncovers different inductive biases of existing methods. The proposed approach consistent gains across several simple NLP tasks for GPT-J (6B) and other small local LMs.

**Strengths:**

1. The paper is well-written and deeply intuitive, easy to follow.
2. The proposed method combined novelty and simplicity. As a researcher who works in closely related areas, I can see myself implementing this and using it as a general framework.
3. The evaluation results are very strong compared to the baselines, except perhaps UDR (Li et al., 2023).
4. The gains transfer better to new LMs and provide explanations, offering a major edge over the baselines.

**Weaknesses:**

1. Until the start of page 4, the paper is very highly opaque on what it proposes! This is surprising given how the criteria proposed (difficulty, coverage, and semantic shift) are actually really intuitive, well-motivated, well-executed, and certainly novel in their usage here. Why not center this clearly?

2. The tasks used for evaluation, and the LM used for evaluation, are deeply underwhelming. There has been a very significant shift in open LLMs since MPT, Llama, Falcon, and LLama2, and many others, particularly Llama2. The same applies to closed LMs, though I am opposed to requiring them. The paper reports several-year old models, namely, GPT-J (6B) and GPT-2 (124M). This helps compare with related work, but it makes the utility of the results somewhat questionable in the near future.

3. How large is the training set? Couldn't cross-validation be enough in these cases with so much data? What's the intuition for not simply training a model to predict good examples, or indeed, just using the "best" frozen set found by sampling and doing cross-validation.

**Questions:**

see weaknesses

---

### Official Review · Reviewer_y4Ji · 2023-11-08

**Soundness:** 2 fair
**Presentation:** 2 fair
**Contribution:** 2 fair
**Rating:** 5
**Confidence:** 3

**Summary:**

This paper mainly focuses on how to effectively select demonstrations for in-context learning (ICL). Prior efforts resort to retrieving demos and then ranking them based on a scoring language model. Ranking results are used to train a dense retriever for demo selection at test time. This paper proposes to include three linguistically-motivated and explainable metrics to assess the efficacy of demo selection. The approach is based on reranking candidates, thereby making it unnecessary to rely on language model-based scoring, while simultaneously facilitating the development of a dense retriever for selecting in-context demonstrations. Experimental results show that the proposed approach outperforms traditional heuristic and dense retrieval methodologies in terms of performance on seven datasets and diverse experimental conditions.

**Strengths:**

1. The idea of the paper is simple and easy to follow.
2. The problem of how to effectively select demos for in-context learning becomes more important in the era of large language models.
3. The proposed approaches (batch and sequential variants) achieve state-of-the-art results on 6 tasks, including sentiment analysis, natural language inference, paraphrase identification and linguistic acceptability.

**Weaknesses:**

1. My main concern is the performance of the proposed method. As shown in Table 1, the proposed method is compared with EPR and UDR. However, the results reported in EPR and UDR [1] are based on GPT-Neo-2.7B [2] as the scoring LM and the inference LM. If we compare the results reported in UDR and LEGEDR in this paper: 92.4 vs 68.2 on SST2, 85.2 vs 65.3 on RT, 78.9 vs 68.8 on CoLA, 65.3 vs 56.3, and 96.6 vs 40.2 on Trec. For another baseline EPR, we also observe a similar large performance gap. I understand that GPT-Neo-2.7B is a larger language model than GPT2 used in this paper. Experimental results based on GPT-Neo-2.7B would be needed for fair comparisons. Moreover, including results on larger models would help to justify if the proposed method (linguistically grounded metrics) still works with more capable language models.
2. It seems to me that these three evaluation metrics introduced in this paper are not well-motivated. The authors claim that these metrics are "linguistically-grounded'. Unfortunately, I did not find any evidence in the paper to support this claim. In section 3 proposed approach, all the motivations behind these three metrics difficulty, coverage and semantic shift are "We hypothesize that a good set of demonstrations should do xxx" without any references or results from preliminary experiments. Based on this, I hardly agree that these three metrics can be defined as "linguistically-grounded'. The metrics proposed are more like "heuristics".
3. This paper claims that existing efforts are "Lack of explainability", as they "do not provide human comprehensible justifications to rationalize their selection process or insights into which demonstration factors contribute to their selection" in the introduction section. They further claimed that their method is explainable. The only evidence provided to support this claim is in Figure 2, which illustrates the normalized contribution of each criterion obtained from demonstrations selected by each model. I am not fully convinced that the proposed method provides the explainability missing in the prior works. It seems to me that this method only justify that the proposed method maintains a better balance on these 3 metrics they introduced than prior efforts.
4. Missing baselines [3]
5. The presentation of the paper still has room for improvement:
5.1 It seems to me that the content in the explanation of the demonstration has nothing to do with the explainability.
5.2 There is an additional "and" in the second main contribution and "Our main contributions include" missed a ":" in the introduction section.

References:

[1] Unified Demonstration Retriever for In-Context Learning. ACL 2023

[2] GPT-Neo: Large Scale Autoregressive Language Modeling with Mesh-Tensorflow.

[3] Compositional Exemplars for In-context Learning. ICML 2023

**Questions:**

See above.